# Unsupervised Detection of Lesions in Brain MRI using constrained adversarial auto-encoders

**Xiaoran Chen**
Computer Vision Lab, ETH Zurich
Zurich, Switzerland
chenx@vision.ee.ethz.ch

**Ender Konukoglu**
Computer Vision Lab, ETH Zurich
Zurich, Switzerland
kender@vision.ee.ethz.ch

## Abstract

Lesion detection in brain Magnetic Resonance Images (MRI) remains a challenging task. State-of-the-art approaches are mostly based on supervised learning making use of large annotated datasets. Human beings, on the other hand, even non-experts, can detect most abnormal lesions after seeing a handful of healthy brain images. Replicating this capability of using prior information on the appearance of healthy brain structure to detect lesions can help computers achieve human level abnormality detection, specifically reducing the need for numerous labeled examples and bettering generalization of previously unseen lesions. To this end, we study detection of lesion regions in an unsupervised manner by learning data distribution of brain MRI of healthy subjects using auto-encoder based methods. We hypothesize that one of the main limitations of the current models is the lack of consistency in latent representation. We propose a simple yet effective constraint that helps mapping of an image bearing lesion close to its corresponding healthy image in the latent space. We use the Human Connectome Project dataset to learn distribution of healthy-appearing brain MRI and report improved detection, in terms of AUC, of the lesions in the BRATS challenge dataset.

## 1 Introduction

Brain lesions refer to tissue abnormalities, which can be caused by various phenomenon, such as trauma, infection, disease and cancer. In the treatment of most lesions, early detection is critical for good prognosis and to prevent severe symptoms before they arise. Medical imaging, in particular, Magnetic Resonance Imaging (MRI), provides the necessary in-vivo observation to this end, and advances in imaging technologies are improving the quality of the observations. What is becoming a bottleneck is that the number of experts who can analyze the images does not grow as fast as the number of patients and images to be studied. Machine learning provides a viable solution to accelerate radiological studies and make detection progress more efficient.

The problem of automatically detecting and segmenting lesions has attracted considerable attention from the research community. Earlier works such as [1],[2] and [3] have suggested effective schemes for lesion detection and segmentation on brain MRI images using different methods. Public challenges, such as The Multi-modal Brain Tumor Image Segmentation (BRATS) and Ischemic Stroke Lesion Segmentation (ISLES), have helped identify several promising methods at the time in the benchmark published in 2013 [4]. Recently, evolution of Convolution Neural Networks (CNN) has provided with advanced supervised neural network solutions, such as [5] and [6], that set the current state-of-the-art.

Success of supervised CNN-based methods is supported by large amount of high-quality annotated datasets. Networks have a large number of free parameters and thus also a large number of examples are needed to avoid over-fitting and yield highly accurate segmentation tools. Contrary to this, human beings can detect most lesions instantly and segment abnormal-looking areas accurately even when

1st Medical Imaging for Deep Learning (MIDL 2018), Amsterdam, Netherlands.

they have not been extensively trained. Having seen couple of healthy brain images provide them the necessary prior information to detect abnormal-looking lesions. We note that this is definitely not the same as *identifying* the lesion (i.e. glioma, meningioma, multiple sclerosis), which is a much more difficult problem.

Algorithms that can mimic humans' ability to detect abnormal-looking areas using prior information on healthy-looking tissue would be extremely interesting. First, they would be essential in developing further methods that require a smaller number of labeled examples to build lesion detection tools and possibly identification tools. Second, such algorithms can easily generalize to previously unseen lesions, similar to what humans can currently do. Lastly, it is an interesting scientific challenge that may be at the heart of the difference between human and machine learning. Motivated by these aspects, we focus on unsupervised detection of lesions by learning the prior distribution of healthy brain images.

Variational Auto-Encoder (VAE) model [7] and models related to it, such as Adversarial Auto-Encoders (AAE) [8] have been successfully used for approximating high-dimensional data distributions of images and outlier detection [9]. They are particularly interesting for unsupervised detection of abnormal lesions because they allow approximating likelihood of a given image with respect to the distribution they were trained on. In their auto-encoder structure, a given test image is first encoded and then decoded, i.e. reconstructed. Both encoded latent representation, assuming outliers lie separate from normal data samples, and the reconstruction are studied for outlier detection [9] in different computer vision tasks.

In this work, we investigate VAE and AAE models for unsupervised detection of lesion in brain MRI. We identify a relevant drawback of these models, lack of consistency in latent space representation, and propose a simple and efficient way to address it by adding a constraint during training that encourages latent space consistency. In our experimental analysis, we train VAE, AAE and AAE with proposed constraint models with T2-weighted healthy brain images extracted from the Human Connectome Project (HCP) dataset. We then use the learned distributions to detect abnormal lesions in an unsupervised manner in the T2-weighted images of the BRATS dataset, where lesions correspond to brain tumors.

## 2   Related works

Similar to supervised learning, deep learning methods have also yielded state-of-the-art methods for approximating high dimensional distributions, especially those related to imaging data. The two main groups of methods are based on Generative Adversarial Networks (GAN) [10] and Variational Auto-Encoder (VAE) [7]. Both GAN and VAE are based on latent variable modeling but they take different approaches to approximate the distribution using a given set of samples. GAN approximates the distribution by learning a generator that can convert random samples from a prior distribution in the latent space to data samples that an optimized classifier cannot distinguish. The obtained data distribution is implicit and GAN is mainly a sampler. Unlike GAN, VAE uses variational inference to approximate the data distribution. It constructs an encoder network that approximates the posterior distribution in the latent space and a decoder that models the likelihood. The probability of each given sample can be directly approximated. Several recent works have built on both GAN and VAE to improve them. [11], [12],[13] have contributed to stabilizing the training of GAN and also enriched the understanding from theoretical aspects. [8], [14] and [15] extend the original VAE model to enable better reconstruction quality and interpretable latent space.

Both GAN-based and VAE-based methods have been applied to abnormality detection, see [9] for a recent review. Example applications include detection of abnormal events in motion [16] and temporal data [17], [18]. More relevant to this article, recent works such as [19] and [20] have proposed to detect abnormal regions in medical images. AnoGAN proposed in [19] trained a GAN on healthy retinal optical coherence tomography images, and later for a given test image, determined the corresponding "healthy" image by performing gradient descent in the latent space. The difference between reconstructed and original image is then used to define an abnormality score for the entire image and the pixel-wise difference is used for detecting the abnormal areas in the images. This approach differs from the VAE-based methods in the way they reconstruct the "healthy" version of the input image. [20] on the other hand, used 3D convolutional auto-encoders to detect abnormal

areas in head CT images. They use reconstruction error as the main metric for abnormality. This work is similar to the basic VAE-based approach.

## 3 Methodology

### 3.1 Generative models

We perform unsupervised anomaly detection in two stages. In the first stage, given a set of healthy images $X_h \in R^{d \times d}$, we train models to learn a distribution $P(X_h)$. Auto-encoder based methods learn this distribution using a latent representation model $P(X_h) = \int P(X_h|z)P(z)dz$, where $z$ is of lower dimension than $X_h$ and $P(z)$ is a predetermined distribution, such as unit Gaussian. These models learn two mappings in the form of networks, namely $Q_\theta(z|X_h)$, mapping high dimensional data $X_h$ to lower dimensional latent representation $z$, and $P_\phi(X_h|z)$, reconstructing the images $X$ encoded in the space of $z$. These two mappings are also known as encoder and decoder.

In the second stage, with the obtained $P(X_h)$, we feed into the models the images that contain abnormal regions, such as lesions. The models trained only with healthy images are not able to reconstruct $X_a$ accurately, indicating low probability with respect to the distribution of healthy images. Abnormal regions are then detected by pixel-wise intensity difference between original image and its reconstruction. Specifically, we performed our analysis using variational auto-encoder and the more recently proposed adversarial auto-encoder.

Proposed in [7], variational auto-encoder (VAE) is based on an auto-encoder structure with latent inference enabled by stochastically sampling in the latent space. The model matches the approximated posterior distribution $Q_\theta(z|X)$ with the prior distribution in the latent space, by minimizing Kullback-Leibler (KL) divergence $D_{KL}(Q_\theta(z|X)||P(z))$. This term acts as a regularization term in addition to $L_2$ reconstruction loss. The principle behind VAE is to optimize the variational lower bound

$$\log P(X) \geq E_{z \sim Q_\theta}[\log P_\phi(X|z)] - D[Q_\theta(z|x)||P(z)], \tag{1}$$

to maximize the data likelihood for the training samples with respect to the network weights $\theta$ and $\phi$

$$\max_{\theta,\phi} E_{z \sim Q_\theta}[\log P_\phi(X|z)] - D[Q_\theta(z|x)||P(z)], \tag{2}$$

Adversarial auto-encoder (AAE) [8] follows a similar encoding-decoding scheme as VAE and yet replaces KL divergence with Jensen-Shannon (JS) divergence estimated by adversarial learning. To impose the prior latent distribution $p(z)$, the model learns an aggregated posterior,

$$q(z) = \int Q(z|x)P(x)dx \tag{3}$$

and uses a GAN to learn this aggregated posterior such that JS divergence between $q(z)$ and $p(z)$ is minimized. As GAN consists of a generator $G$ and a discriminator $D$, here in AAE, the generator is the encoder $Q(z|X)$. The encoder is optimized to generate latent variables $z \sim q(z)$ that confuses $D$ with $z \sim p(z)$, which is sampled from the prior distribution. On the other hand, $D$ tries to distinguish $z \sim q(z)$ from $z \sim p(z)$. This introduces a $\min \max$ optimization problem. According to [10], the $\min \max$ optimization can be expressed as,

$$\min_G \max_D E_{z \sim q(z|X)}[\log D(z)] + E_{z_{prior} \sim p(z_{prior})}[\log(1 - D(G(z)))] \tag{4}$$

For stable training of GAN, we substituted GAN in the original paper with Wasserstein GAN with gradient penalty (WGAN-GP). Based on WGAN-GP, the optimization problem can be then rewritten as,

$$\min_G \max_D [E_{z \sim q(z|X)}D(z) - E_{z_{prior} \sim p(z)}D(z_{prior})] + \lambda E_{z \sim q(z|X)}[(||\nabla_z(D(z))||_2 - 1)^2] \tag{5}$$

The advantage of AAE over VAE is the coverage in the latent space[8]. Defining $q(z) = \int q(z|x)p(x)dx$, AAE enforces a better match between $q(z)$ and $p(z)$. As a result, any sample from $p(z)$ has a higher chance to be similar to a data sample.

For the decoder, we assume $p(X_h|z) \sim \mathcal{N}(X; \mu(z|X_h), \mathbf{I})$. Accordingly, we choose $L_2$ loss, $||X - X'||_2$ as the reconstruction loss. To generalize, the objective of both VAE and AAE can be written as $L_{auto-encoder} + L_{divergence}$. We implement a ResNet structure for the encoder and decoder.

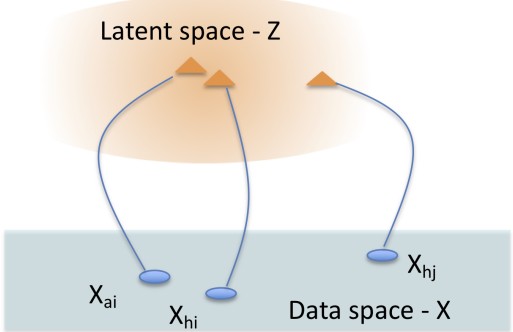
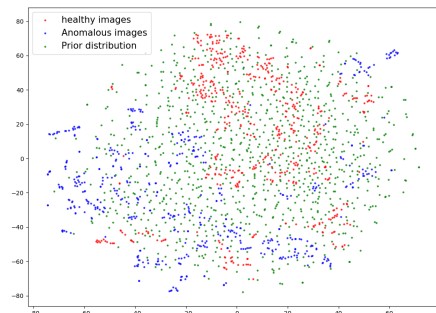

**Figure 1:** Encoding input data into latent representation. Left: Illustration shows the encoding of VAE/AAE model. Data samples (blue circles) are encoded into latent representation (orange triangles). Difference between two 'healthy' images $X_{hi}$ and $X_{hj}$ might be larger than the distance between an image with an abnormal lesion $X_{ai}$ and its 'healthy' version $X_{hi}$. Latent representation of these images may also satisfy the same relationship. As a result, images with abnormalities may not lie separate than normal images. Right: Image shows TSNE embeddings of healthy images (red) and lesion images (blue) from the BRATS dataset. We also show samples from the prior distribution in green. Healthy images and abnormal images can be mapped close in the latent space, the latent representations of the abnormal images also lie in the prior distribution together with the representations of healthy images, making them indistinguishable in the latent space.

### 3.2 Latent representation of brain MRI with abnormal lesions

It would be desirable if the latent representation of abnormal images lie separate than those of normal images. In [19], the authors show results suggesting this behavior for their application. For brain MRI however, the situation can be different due to possibly higher variability across images of healthy brains compared to retinal images. In the case of high variability, the intensity differences caused by abnormal lesion, might be smaller than the differences due to normal variability of brain MRI. For instance, we can observe high variation in intensity when we consider different slices of a 3D volume as can be seen in the different columns in Figure 2.

Technically, suppose we have an image with an abnormal lesion $X_{ai}$ and also a healthy version of the same image $X_{hi}$, i.e. without the lesion. The lesions that we aim to detect are local intensity variations that results in a certain distance in the image space: $||X_{ai} - X_{hi}||_d$. Depending on this distance value, there might be other healthy images with $||X_{hj} - X_{hi}||_d > ||X_{ai} - X_{hi}||_d$ for both $d = 2$ as used here or $d = 1$ as used in [19]. Given that both encoding and decoding are continuous functions, the latent representation of $X_{ai}$ can be closer to $X_{hi}$ than $X_{hi}$ to $X_{hj}$. If the projection of $X_{hi}$ lies within the center of the prior distribution in the latent space, then the projection of $X_{ai}$ may also lie within the center, hence not separable. We display this behavior using TSNE visualization for the HCP and BRATS datasets.

### 3.3 Imposing representation consistency in the latent space

As explained above, the abnormal images are not necessarily mapped outside the predetermined latent distribution. However, they can still be detected by evaluating the residual image $||X_{ai} - X'_{ai}||$, where $X'_{ai}$ is the model reconstruction of $X_{ai}$. If we can assume that an image with a lesion will map to a very similar point in the latent space as the same image without the lesion, then the residual image would highlight the lesion section. In the ideal setting, we would enforce this with a paired dataset. However, we do not have access to such datasets during training. Instead, we have access to the images of healthy subjects $X_h$ and we can get their reconstructed versions $X'_h$. As a proxy to the ideal case, we propose to enforce consistency between the latent representations of these images.

We impose the consistency in the latent representation by adding a regularization term $||z_h - z'_h||^2$ in the auto-encoder loss, where $z$ is the projection of the healthy image $X_h$ and $z'_h$ is the projection of

the reconstruction $X'_h$. As a result the auto-encoder loss becomes $L_{auto-encoder} = ||X_h - X'_h||^2 + \lambda||z_h - z'_h||^2$, where $\lambda$ controls the weight of the new regularization term. If $\lambda \to 0$, the objective remains the same as the original objective function; if $\lambda \to \infty$, the model ignores $L_2$. Thus $\lambda$ serves to control how similar the images are so that they can be mapped close in the latent space.

## 4  Datasets

We use Human Connectome Project (HCP) T2-weighted structural images as our training data. The dataset contains images from 35 healthy subjects. As the test data, we use T2-weighted images of 42 subjects from the Multimodal Brain Tumor Image Segmentation (BRATS) Challenge 2015 and perform lesion detection on these images.

**Bias correction** We perform bias correction for BRATS challenge dataset using n4ITK bias correction /citetustison2010n4itk.

**Data preprocessing** We normalize the histograms by subject for both HCP and BRATS datasets such that they follow the same histogram profile. Data is also standardized to have zero mean and unit variance.

As reconstructing large images of high resolution is a challenging topic and our work does not discuss methods to improve reconstruction quality, we down-sampled original images to the size of 32×32 so that VAE and AAE are able to reconstruct them with satisfactory quality.

## 5  Results

We experiment with two different $\lambda$ values within the AAE model, 0.5 and 1, and compare detections with the VAE and AAE models. We chose to use the AAE model to experiment with due to its theoretically better behavior in the latent space.

### 5.1  Reconstructing healthy images

In order to learn the distribution of healthy data, we train the models to reconstruct the input images and at the same time to minimize their respective divergences. While $Q(z|X)$ matches $p(z)$, quality of reconstruction indicates how well the data distribution is captured. First, to illustrate that models' capabilities, we reconstructed healthy images of test data from HCP datasets. Results are shown in Figure 2.

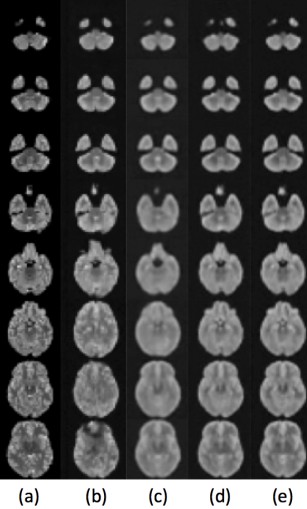

(a)    (b)    (c)    (d)    (e)

**Figure 2:** Reconstruction of a healthy image. Column (a) shows a T2w images from a test subject from the HCP dataset; Columns (b) to (e) are reconstruction by VAE, AAE, AAE with $\lambda = 0.5$, and AAE with $\lambda = 1.0$.

## 5.2 Improved detection with latent constraint

Then we move on to detect lesions using the healthy distribution learned with those models. Images with anomalies from BRATS datasets are used for anomaly detection. Lesions detected using the four models are presented in Figure 3. Residual images are computed as pixel-wise absolute intensity difference between input anomalous image $X_a$ and its reconstruction $X_a'$ as $|X_a - X_a'|$.

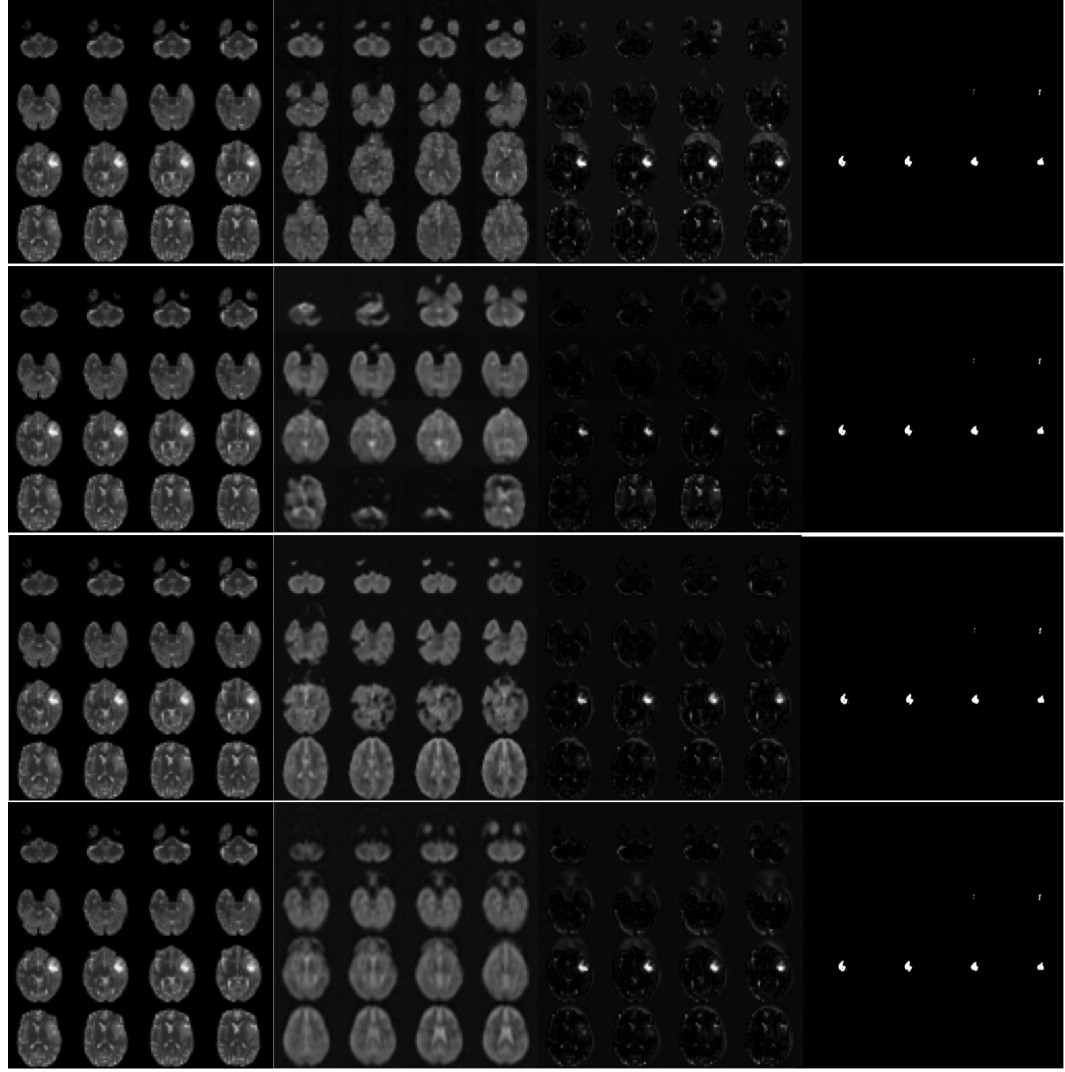

**Figure 3:** Unsupervised lesion detections through reconstruction error. Each row of 4 images from top to bottom show results by VAE, AAE, AAE with $\lambda = 0.5$, AAE with $\lambda = 1.0$. Each column of 4 images show respectively, the original BRATS T2w images, reconstruction of the original images, residual images between the original images and the reconstructed images, the ground truth lesions visible in T2w images available with the BRATS dataset.

Each of the models exhibit capability to detect lesions by computing residuals images, although their performance differ from one another. In the reconstructed images, the models are not able reconstruct abnormal regions and fills in healthy tissue within the lesion area. This indicates that the learned data distribution excludes such samples and supports the reconstruction based approach to anomaly detection. Without latent constraint, the reconstructed images show larger differences to the image with abnormality in areas other than the lesion itself. In the case of reconstruction with VAE, the reconstruction of abnormal images tends to be less sharp although the overall appearances are preserved. In comparison with VAE, AAE produces sharper reconstruction. However, reconstructions of abnormal images appear to be unrealistic and do not preserve the shape of valid brains. Both

reconstruction and detection are shown to be improved with latent constraints. With the latent consistency imposed in AAE, the model is able to output reconstructions that look more consistent with the input in the healthy region. This constraint also ensures that the reconstructions preserve realistic appearances of a brain. The differences in the residual images are mostly highlighting the lesion areas.

We note that it is also necessary to choose a proper $\lambda$ for the specific dataset to obtain satisfactory results. We compare the lesions detected with $\lambda = 1.0$ and $\lambda = 0.5$. When $\lambda = 1.0$, the model is able to detect the lesions more accurately compared to the other models. When the constraint is relaxed with $\lambda = 0.5$, the lesions are detected with more false positives and some reconstructed images tend to have unrealistic appearances. We plot Receiver Operating Characteristic (ROC) curve

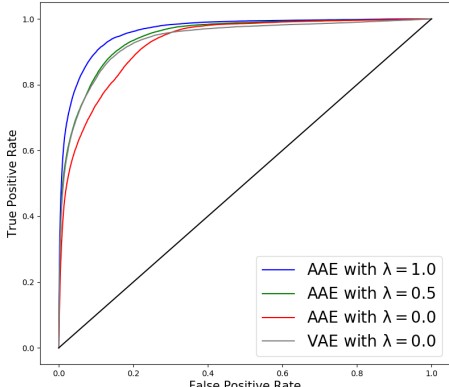

**Figure 4:** Accuracy estimation for pixel-wise anomaly detection performance. ROC curves are calculated based on the four models using anomalous images in the modality of T2 from BRATS datasets. AAE and VAE models with $\lambda = 0$ corresponds to the original algorithms.

as in Figure 4 to quantitatively evaluate detection ability of different models. The ROC curves are computed comparing the residual images with the ground truth segmentations for the T2w images. Among the models, AAE with $\lambda = 1.0$ achieved the highest Area Under Curve (AUC), which accords with the detection performance shown in Figure 3. Although VAE produces blurry images compared to other models, this drawback does not significantly impair its ability to detect lesions.

**Table 1:** AUC are calculated for Figure 4 for the listed models

|  | Models | | | |
| --- | --- | --- | --- | --- |
|  | VAE | AAE | AAE, $\lambda = 0.5$ | AAE, $\lambda = 1.0$ |
| **AUC** | 0.897 | 0.885 | 0.906 | 0.923 |

In Figure 5, we show the distributions of pixel-wise reconstruction errors for healthy tissue and lesions in the images from the BRATS dataset. Details of each distribution are summarized in Table 2. We show normalized histograms as well as Gaussian fits. For successful algorithm, we would expect the error distribution of lesion pixels to be separated from that of healthy pixels. Figures suggest a larger separation for $\lambda = 1$. To quantify this, we also estimate the overlapping area between the error distribution of healthy and anomalous pixels. The overlap is calculated as the number of anomalous pixels that have errors lying inside 95% confidence interval for the error distribution of healthy pixels. This indicates that such anomalous pixels cannot be detected in terms of statistical measures and will appear as false negatives. Smaller overlap indicates further separation between the distributions and therefore more accurate detection. The overlapping regions given by the models are related to their AUC values. VAE, AAE ($\lambda = 0.0$) and AAE ($\lambda = 0.5$) exhibit similar overlapping areas, where AAE ($\lambda = 0.5$) is slightly better than the other two. With effective latent constraints, AAE ($\lambda = 1.0$) show the least overlap among the models.

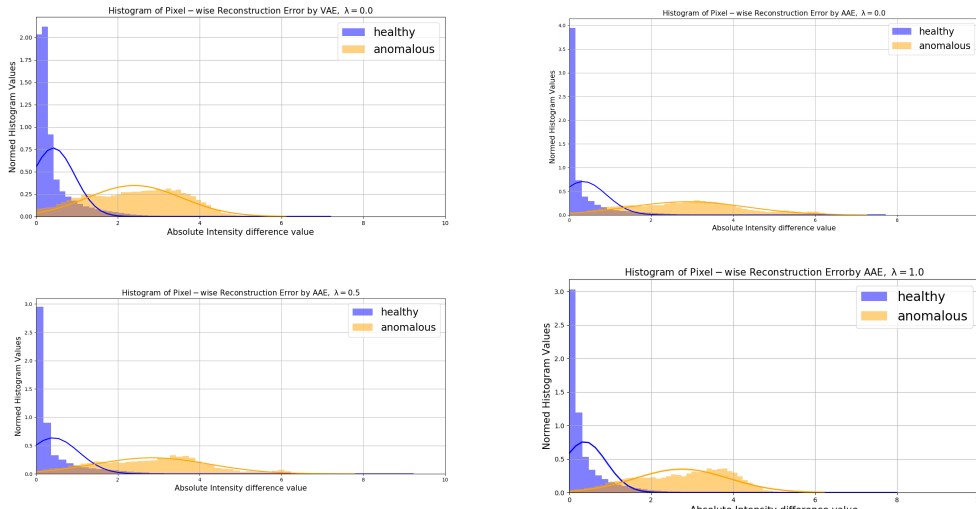

**Figure 5:** Histograms are calculated to show pixel-wise reconstruction loss in the form of absolute intensity difference. The histograms are normalized. We also show Gaussian fits for the histograms. The normalization ensures unit integral hence, y-axis values can be larger than 1.

**Table 2:** $\mu$ and $\sigma$ are calculated for the distributions in Figure 5. $\mu_h$ and $\sigma_h$ denote mean and standard deviation for healthy distribution, $\mu_a$ and $\sigma_a$ denote that of anomalous distribution.

| Models | $\mu_h$ | $\sigma_h$ | $\mu_a$ | $\sigma_a$ | Overlapping (%) |
|---|---|---|---|---|---|
| VAE, $\lambda = 0.0$ | 0.424 | 0.521 | 2.405 | 1.152 | 26 |
| AAE, $\lambda = 0.0$ | 0.343 | 0.565 | 2.894 | 1.431 | 28 |
| AAE, $\lambda = 0.5$ | 0.422 | 0.625 | 2.834 | 1.401 | 25 |
| AAE, $\lambda = 1.0$ | 0.379 | 0.526 | 2.750 | 1.145 | 17 |

# 6 Conclusion

In this work, we approached the challenge of lesion detection in an unsupervised-learning manner by learning prior knowledge from healthy data and detect abnormalities according to the learned healthy data distribution. We investigated the detection performance abnormality based methods, namely VAE and AAE using brain MRI images. We then analyzed the behavior of these models and proposed a latent constraint to ensure latent consistency and enable more accurate detection of abnormal regions. We showed that the abnormal lesions can be detected with the investigated models and the accuracy of detection can be improved with our proposed latent constraint. A natural competitive to the models we presented is the AnoGAN model. At the time of submission, although we could train the necessary DCGAN, we were not able to get decent results from AnoGAN due to problems in the gradient decent in the latent space despite all our efforts. Consequently, we refrain from show those results and keep this comparison for future work.

## Acknowledgements

This work is partially supported by Swiss National Science Foundation.

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
