# OpenReview forum: "Unsupervised Detection of Lesions in Brain MRI using constrained adversarial auto-encoders"
_MIDL.amsterdam/2018/Conference — MIDL 2018 Poster_

### Review · AnonReviewer1 · 2018-05-08
**ready for prime time ?**

**Rating:** 3
**Confidence:** 2

**Review:**

The authors learn the structure of brain MRI using variational auto-encoders. They introduce a term in the training  that enforces similarity of scan and reconstructed scan in image space, as well as in the latent space.

Pro: Adding the constraint is an intuitive idea and results are promising. Results on brain lesion MRI show good results in delineating the abnormal areas.

Con: Tumors are large and residual images appear to be smooth. A harder (probably too hard?) problem would be the identification of MS lesions. Some numerical problems remain.


**Special Issue:**

No

---

### Review · AnonReviewer3 · 2018-05-09
**Good idea, but experiments should be improved**

**Rating:** 2
**Confidence:** 2

**Review:**

The paper proposes to perform brain lesion detection by formulating the problem as anomaly detection. They propose to train an adversarial autoencoder, where the main contribution lies in the  addition of a constraint to the loss function.

Pros:

The paper tries to address an important gap in the literature.

Cons:

The proposed methodology is not assessed with the state-of-the-art for anomaly detection: AnoGan[19].
Results in Table 1 attempt to show the validity of this approach to improve Adversarial autoencoder. Authors also state that no comparison against AnoGan is presented because they could not achieve "decent" results.
However, this lack of results based on AnoGan is a major drawback (because it is the state-of-the-art for anomaly detection),especially given the big difference in AUC between AnoGAN and adversarial autoencoder in [19] (0.89 vs 0.73).

Another concern regarding experiments is the data used to perform the experiments. Images in Figure 3 suggest that it may be possible to detect anomalies by thresholding the image. Could detection results from thresholding be added?

Literature review Sec..2 addresses abnormality detection in medical image analysis, mentions [19,20] and what they propose, but their limitations are not explained there. In contrast, it is Sec. 3.2 (Methodology) explaining limitations.

Clarity and organisation of the paper needs to improved as it is difficult to follow.


Other comments:

I am not convinced that the right word to use in the title is "Unsupervised". In my opinion, it is more appropriate to include "anomaly", which refers to training an algorithm with "normal" cases and try to detect those that do not belong to the distribution of "normal".

In my opinion, the lack of experiments involving the state-of-the-art [19] limits the assessment of the methodology. Therefore, the paper should be improved to be accepted.

**Special Issue:**

No

---

### Review · AnonReviewer2 · 2018-05-09
**Unsupervised Detection of Lesions in Brain MRI using constrained adversarial auto-encoders**

**Rating:** 3
**Confidence:** 3

**Review:**

This paper tackles the problem of detecting lesions in brain MRI scans by learning a representation of healthy cases.
In this way, lesions are detected as outliers outside the manifold of the learned representation.
This approach allows to take advantage of a large set of unlabeled data of healthy cases without the needs for manual annotations. For his purpose, unsupervised deep learning approaches are used, namely Variational Auto Encoders (VAE) and Adversarial Auto Encoders (AAE).

The authors propose to modify the loss function of the auto encoder by adding a regularization data term, which enforces similarity between latent representation of input images and of corresponding reconstructed images.
Experiments on the publicly available BRATS dataset show that detection using VAE is superior to AAE when the extra loss term is not used, but that AAE outperforms VAE when the additional term is used.

The quality of this paper is overall good, the idea of learning only from healthy cases is interesting, although not completely novel, and the effect of the used additional term in the loss seems effective.

However, there are several aspects in the paper that should be addressed to improve its clarity.
* Values of lambda = 0.5 and lambda = 1 are used. The authors should report in the paper why they only experimented with this two values.
* The size of the latent space is not reported.
* In Figure 3, the caption refers to Brats for details on the ground truth. It would be useful to indicate the case-id, to quickly identify cases in Brats. It is also clear from Figure 3 that the reconstructions using different methods are different but in the end the difference with the input image, which determines lesion detection, looks very similar. It would be interesting to comment more on the benefit of each method and their difference when it comes to lesion detection.
* ROC analysis: it is not clear how the ROC analysis is performed. The paper says "The ROC curves are computed comparing the residual images with the ground truth segmentation". Are voxels in the residual used in the ROC analyis?

**Special Issue:**

No

---

### Decision · Program_Chairs · 2018-05-15
**Paper115 Acceptance Decision**

Poster